# Task-oriented training in stroke rehabilitation: Qualitative study on perspectives and challenges among Pakistani physiotherapists

**Ayesha Afridi** **\*, Arshad Nawaz Malik, Neha Raheel, Umme Aiman, Fajar Shahid, Aneeqa Iftikhar, Maryam Rehman**

Rehabilitation Department, Riphah International University, Islamabad, Pakistan

\* afridi.ayesha@gmail.com

## Abstract

### Objective

The aim of the study is to explore how Pakistani physiotherapists integrate Task Oriented Training (TOT) in stroke rehabilitation, categorizing its use across motor function, cognitive rehabilitation, and balance training, while identifying barriers and adaptation strategies.

### Methods

A qualitative descriptive study was conducted using semi-structured interviews with 22 experienced physiotherapists. Data were analyzed using Braun and Clarke's six-step thematic analysis, with ATLAS.ti9 for coding.

### Results

Among the 22 participating physiotherapists, 18 (82%) reported incorporating TOT for motor and balance recovery, while only 8 (36%) emphasized cognitive rehabilitation due to training gaps and patient engagement challenges. Major barriers included resource limitations reported by 15 participants (68%), time constraints by 12 participants (55%), and cultural beliefs by 12 participants (55%), which therapists addressed through home-based modifications and caregiver involvement.

### Conclusion

TOT is a valuable yet underutilized approach in Pakistani stroke rehabilitation, requiring standardized guidelines, cognitive rehabilitation training, and policy reforms to enhance effectiveness. Future research should focus on cost-effective, scalable rehabilitation models to improve stroke recovery outcomes in resource-limited settings.

**Data availability statement:** "All relevant data are within the paper and its Supporting Information files."

**Funding:** The author(s) received no specific funding for this work.

**Competing interests:** The authors have declared that no competing interests exist.

## Introduction

Stroke survivors often have complications including communication and physical impairments, influences survivors' ability to resume work, leading to a long-term decline in their quality of life and that of their families [1], commonly affects relatively young adults, with a mean age of approximately 42 ± 12.6 years. [2] In Pakistan, stroke poses a significant public health burden, with an age- and sex-standardized incidence rate of 131.6 per 100,000 population (95% CI: 121.7–142.6). The stroke mortality rate is similarly high, recorded at 133.7 per 100,000 (95% CI: 112.7–155.7), resulting in an incidence-to-mortality ratio (IMR) of 0.98, indicating a substantial fatality burden. The DALYs lost due to stroke are estimated to be 2,534 per 100,000 (95% CI: 2,117.5–2,947.6), highlighting the long-term consequences of stroke-related disabilities in the country. [3] The associated economic burden on stroke survivors in these regions is also expected to increase. These consequences highlights the need of effective rehabilitation strategies that enhance functional independence and quality of life [4].

Among the various rehabilitation approaches is task-oriented training (TOT), based on the systems theory of motor control, focuses on task-specific strategies to enhance adaptability in neurological rehabilitation. It promotes skill practice to improve problem-solving, environmental adaptation, and reward-based learning. [5] Literature suggests that TOT improves motor control, coordination, balance, and cognitive skills by improving neuroplasticity based recovery. [6] However, while TOT has been extensively considered in Western contexts of rehabilitation, [7] its application in LMICs, particularly in Pakistan, remains under explored and some limitations in incorporating TOT in Pakistan were also reported in mini-review [8].

Pakistani physiotherapists work within a specific socio-cultural and healthcare framework that is different from already existing western rehabilitation models. [9] The accessibility, acceptance, and execution of rehabilitation strategies are primarily based on societies' cultural norms, healthcare system, and socioeconomic factors. [10] High-income countries have often patient-centered and technology-driven rehabilitation services, [11] while Pakistani rehabilitation system is influenced by financial constraints, traditional beliefs and practices, and inadequate access to specialized care. [12] Furthermore, exposure to evidence-based practice (EBP) among physiotherapists, [13] different educational backgrounds, [14] and resource constraints [15] may affect how TOT is implemented in clinical settings. Understanding the TOT strategy within Pakistan's rehabilitation landscape is important for developing culturally relevant, feasible, and effective rehabilitation protocols.

While previous studies have extensively reported the benefits of TOT in stroke rehabilitation [16,17] most studies have focused on standardized protocols developed in high tech and advanced healthcare settings. [18,19] This may lead to a limited understanding of how TOT is adapted and practiced in various healthcare systems. Specifically, there is a lack of qualitative study about experiences, challenges, and adaptations when integrating TOT into their rehabilitation practices among Pakistani physiotherapists.

To address this knowledge gap, this study is based on qualitative descriptive approach to explore how Pakistani physiotherapists integrate TOT into stroke rehabilitation. A qualitative method is mainly suited to this study because this approach is well-suited for exploring complex constructs holistically, providing deep insights into lived experiences across diverse populations. [20] Qualitative research may also captures the lived experiences of rehabilitation professionals, shedding light on the practical adaptations, barriers, and facilitators [21] influencing TOT implementation in stroke rehabilitation. This study aims to explore that how physiotherapists conceptualize, modify, and apply TOT within limited resources of Pakistan's healthcare system using semi-structured interviews and thematic analysis.

## Materials and methods

### Research design

Qualitative descriptive design and a thematic framework was used in this study to explore TOT experiences, practices, and perspectives of physiotherapists in Pakistan. This approach captures detailed narratives and complex experiences. A qualitative descriptive design was selected because the main aim was to provide a comprehensive, data-driven account of current TOT practices with straightforward representation of participant perspectives. A thematic framework approach guided data analysis, allowing for the systematic identification, organization, and interpretation of key themes emerging from the data. [22] Specifically, Braun and Clarke's six-step thematic analysis [23] was used to ensure a structured and rigorous process of data organization, coding, and interpretation. This study adhered to the Consolidated Criteria for Reporting Qualitative Research (COREQ) guidelines.

### Sample size and participants characteristics

A sample of 22 physiotherapists was recruited based on purposive sampling, ensuring that all participants had at least five years of clinical experience in stroke rehabilitation and held a Master's degree in Neuro-Muscular Physical Therapy. The inclusion criterion of a Master's degree in Neuro-Muscular Physical Therapy was selected to ensure a homogeneous sample of physiotherapists with advanced training and specialization in stroke rehabilitation. This qualification reflects the minimum advanced educational standard for neuro-rehabilitation expertise in Pakistan and was intended to ensure that participants possessed sufficient theoretical and clinical insight into the application of task-oriented training in neuro-muscular disorders.

Participants were recruited through professional physiotherapy associations, academic networks, and direct referrals from rehabilitation centers in urban and semi-urban areas of Pakistan. The recruitment period for this study was from September 15, 2022, to October 25, 2022. The sample size was determined using data saturation as a guiding principle, where no new themes or variations in responses emerged from additional interviews, indicating a comprehensive exploration of the phenomenon [24]. To document saturation, a saturation table was maintained, (S3 in Supporting information) systematically tracking themes, sub-themes, and their recurrence across interviews. Saturation was assessed inductively with emerging codes refined iteratively throughout the analysis. No additional participants were recruited once thematic redundancy was reached in the last two interviews.

### Data collection procedure

To minimize interviewer bias, four trained female researchers (UA, MR, AI and NR) with a Doctor of Physical Therapy degree conducted the interviews. Training emphasized unbiased questioning techniques, active listening, and neutral prompting to avoid leading responses. Additionally, standardized semi-structured interview protocols were used to ensure consistency across interviews. A total of 22 interviews were conducted, held face-to-face in their rehabilitation clinics. Privacy was ensured, and all sessions were audio-recorded with participant consent for accurate transcription and analysis. Non-verbal cues were carefully recorded in the notes.

Power dynamics between interviewers and participants were carefully managed by:

- Conducting interviews in neutral, private settings to reduce hierarchical influences.
- Encouraging participants to speak freely and critically, ensuring they felt comfortable sharing both positive and negative experiences.
- Using non-verbal rapport-building techniques to create a conversational rather than interrogative environment.

### Reflexivity discussion

Given that the interviewers were physiotherapists, their professional backgrounds could have influenced the way questions were framed and how participants responded. To address researcher position, reflexive memos were maintained throughout data collection and analysis, documenting personal biases, assumptions, and reflections on participant responses. Researchers maintained reflexivity by cross-checking interpretations with an external expert to minimize personal biases.

The interview format was based on participant preference and logistics, maintaining consistency through standardized protocols and questions. The semi-structured interview guide was validated by three independent expert physiotherapists each holding a Master's degree in Neuro-Muscular Physical Therapy and with ≥7 years of clinical experience in stroke rehabilitation, including prior use of task-oriented training (TOT). These experts reviewed the tool for clarity, relevance, and cultural appropriateness, and confirmed that all items effectively assess clinicians' familiarity with and application of TOT. (S1 **in Supporting information-** S2 **in Structured Interview Questionnaire)**

Before data collection, a pilot test was conducted to ensure the interview guide was clear, perceivable, and effectively addressed the research questions. The questionnaire was drafted and administered in English, the medium of instruction and clinical communication for all participants. Where necessary, interviewers provided Urdu explanations for technical terms to ensure comprehension. This pilot test involved three physical therapists with five years of clinical experience from September, 1st to 10th, 2022. Piloting of interview guide leads to minor adjustments in wording for clarity and precision. Questions covered TOT activities for post-stroke recovery, including upper limb, lower limb, balance, and cognition, as well as cultural/contextual factors and barriers. Each interview lasted approximately 50–60 minutes.

### Member checking & credibility

To enhance trustworthiness, member checking was employed at two stages:

1. During interviews – Key responses were paraphrased and confirmed with participants in real time to ensure accurate interpretation.
2. Post-analysis – A summary of thematic findings was shared with six randomly selected participants to validate whether the identified themes accurately represented their experiences. Minor refinements were made based on their feedback.

Additionally, triangulation was employed by comparing findings across different geographic locations, experience levels, and institutional settings to ensure a well-rounded perspective.

### Data analysis

Data were analyzed using Braun and Clarke's six-step thematic analysis [25], involving:

1. Familiarization with data – Transcripts were read multiple times for immersion.
2. Initial coding – Line-by-line coding was conducted by five independent coders using an inductive approach.
3. Theme identification – Codes were grouped into emerging patterns.

4. Theme review and refinement – Inter-coder reliability was ensured through consensus discussions. Disagreements were resolved via majority voting or secondary review by an external qualitative expert.

5. Defining themes – Finalized themes were reviewed in relation to research objectives.

6. Report writing – A thematic structure was developed.

**Use of ATLAS.ti9 for thematic coding**

ATLAS.ti9 was used to facilitate systematic data organization, retrieval, and coding consistency [26]. Specifically:

• Coding was software-assisted, but final decisions were made manually to preserve contextual depth.

• The software was used to visualize code relationships, compare participant responses, and track co-occurring themes.

**Ethical considerations**

Researchers received training in qualitative research ethics, and the study was approved by the ethics committee. There were no pre-existing relationships between the researchers and participants before the study. Participants were informed about the study's purpose and provided written informed consent before the interviews. They were also informed about the researchers' qualifications and the study's background during the consent process, but personal goals and motivations were not disclosed to prevent response bias.

Ethical approval: Riphah International University ethics committee Riphah/RCRS/REC/01424).

## Results

### Participants

A total of 40 physiotherapists were invited to participate, of whom 22 physiotherapists met the inclusion criteria and consented to interviews. The participants had a mean clinical experience of 5.45 ± 0.59 years and an average age of 35.09 ± 2.81 years. The sample was predominantly male (68.2%), with 11 participants (50%) based in Lahore, followed by Faisalabad (22.7%), and smaller representation from cities like Peshawar, Rawalpindi, Jhelum, and Islamabad (Table 1).

### Themes

Analysis of the interview data revealed six overarching themes, each comprising multiple sub-themes. Braun and Clarke's thematic analysis was used to derive these themes inductively, and negative cases were specifically sought to ensure

**Table 1. Demographic characteristics of participants.**

| Demographics | | Frequencies n (%) |
|---|---|---|
| **Gender** | Male | n = 15 (68.2%) |
| | Female | n = 7 (31.8%) |
| **Cities** | Peshawar | n = 2 (9.1%) |
| | Lahore | n = 11 (50.0%) |
| | Islamabad | n = 2 (9.1%) |
| | Rawalpindi | n = 1 (4.5%) |
| | Jhelum | n = 1 (4.5%) |
| | Faisalabad | n = 5 (22.7%) |
| **Age (Years)** | 35.09 ± 2.81 (Mean ± Std. Deviation) | |
| **Years of Experience** | 5.45 ± 0.59 (Mean ± Std. Deviation) | |

analytical depth. All 22 participants (100%) responded to each domain of the semi-structured guide (demographics; understanding and application of TOT; motor, cognitive, and balance training; barriers; cultural factors). The frequency of thematic responses is presented in supporting files. The themes include:

1. **Task-Oriented Training (TOT) Practices**

2. **Motor Function Improvement (Upper & Lower Limbs)**

3. **Cognitive Rehabilitation**

4. **Balance Training**

5. **Challenges and Barriers in Implementing TOT**

6. **Cultural and contextual factors**

1. Task-oriented training (TOT) practices

Majority of the participants 18 (82%) described TOT as a patient-centered approach integrating meaningful and functional activities into rehabilitation. They emphasized the need for functional relevance, patient engagement, goal-oriented approaches, holistic rehabilitation, and adaptability.

**As one participant expressed**

"TOT is not just about repetitive exercises. It's about ensuring that patients can reintegrate into daily activities—reaching for a cup, standing from a chair, or walking on different surfaces." (P8)

However, some physiotherapists expressed concerns about the lack of structured TOT protocols in Pakistan, leading to inconsistencies in practice. As one participant noted:

*"There's no standardized way we implement TOT here. Everyone adapts it differently, depending on their training and resources available."* (P12)

2. Motor function improvement for upper and lower extremities

Most of the physiotherapists 18 (82%) categorized motor function rehabilitation into upper and lower limb training, emphasizing both gross and fine motor skill enhancement.

**Upper limb rehabilitation**

Gross Motor Tasks: *"We focus on larger movements first—reaching, lifting objects, and weight-bearing on the affected limb."* (P5)

Fine Motor Control: *"For fine motor skills, we use object manipulation exercises, like buttoning a shirt or grasping small objects."* (P9)

**Lower limb rehabilitation**

Stepping & Gait Training: Physiotherapists incorporated stepping exercises on uneven terrain and stair climbing to mimic real-world walking conditions.

Negative Case: Some therapists reported limited patient adherence to lower limb training, particularly when early ambulation caused fear of falling.

*"Many patients are hesitant to walk, fearing another stroke or fall."* (P14)

3. Cognitive rehabilitation

In current study, only 8 (36%) participants highlighted the critical role of TOT rehabilitation in enhancing cognitive functions among stroke survivors. This theme encompasses a range of activities and exercises tailored to improve specific cognitive skills. TOT was also used to enhance cognitive functions in stroke survivors, particularly attention, problem-solving, and memory.

- **Attention Training:** 7 (32%) therapists incorporated dual-task exercises that required patients to engage in mental calculations while walking.
- **Quantitative Reasoning:** *"We sometimes use real-life tasks, like handling money, to improve reasoning."* (P3)
- **Memory Retention:** *"We introduce games that require recalling names of objects or sequences."* (P17)

**Negative case: Challenges in cognitive engagement**

Despite the benefits, some therapists faced challenges in engaging patients with cognitive impairments, with low motivation and frustration reported as key barriers:

*"Patients with cognitive deficits often struggle with task adherence. They get frustrated easily."* (P7)

Adaptation Strategy: Therapists found that breaking down tasks into smaller components and using visual prompts improved participation.

4. Balance training

Most of the participants 18 (82%) mentioned importance of balance training in integrating while giving task oriented training. Balance rehabilitation was divided into static and dynamic exercises.

- Static Balance: Included single-leg stands and tandem stance training.
- Dynamic Balance: Included walking on uneven surfaces, obstacle navigation, and perturbation training.

**Negative case: Resource Constraints in balance training**

Many therapists cited the lack of specialized balance equipment as a limitation:

*"We don't always have access to balance boards or VR-based training, so we improvise with basic exercises."* (P11)

**Adaptation strategy.** Some clinics substituted traditional balance tools with household items (e.g., stepping over cushions or walking on textured surfaces).

5. Challenges and barriers in implementing task-oriented training

Current study explored the barriers that physiotherapists encounter when implementing TOT in stroke rehabilitation. Participants shed light on the challenges they face, highlighting four distinct sub-themes. 15 of 22 physiotherapists (68%)

reported resource constraints as a primary barrier. Time constraints were noted by 12 participants (55%). Knowledge gaps and lack of standardized training were identified by 10 therapists (45%). Similarly, 10 participants (45%) described patient-related barriers.

## Resource constraints

• Limited access to TOT-specific equipment and funding shortages hindered implementation.

Negative Case: Some therapists reported that patients lacked access to rehabilitation facilities in rural areas, leading to sub optimal home-based rehabilitation:

*"Many patients discontinue therapy because they can't afford frequent travel to rehab centers."* (P16)

## Time constraints

• Therapists struggled to balance patient loads with the time-intensive nature of TOT.

*"TOT requires more time per session, but we have limited slots for each patient."* (P10)

## Knowledge gaps & standardization issues

• Some therapists lacked formal TOT training, leading to variability in application.

Proposed Solution: Participants suggested structured TOT workshops to standardize implementation across clinics.

## Patient-related barriers

• Cognitive impairments, fear of movement, and low motivation were major challenges.

Negative Case: A participant shared

*"Patients sometimes don't see immediate results, so they lose motivation quickly."* (P20)

Adaptation Strategy: Physiotherapists found that goal-setting, family involvement, and real-life task demonstration improved adherence.

6. Cultural and contextual factors

Current study explored the influence of cultural and contextual factors on the implementation of TOT in stroke rehabilitation. Participants highlighted several key factors that impact the process, encompassing four distinct sub themes: Cultural norms and values were highlighted by 12 therapists (55%). Healthcare system factors, including insurance limitations and urban-centric services, were cited by 11 participants (50%). Socioeconomic factors, particularly work commitments and financial constraints, were reported by 10 therapists (45%). Finally, language and communication barriers were mentioned by 7 participants (32%), who emphasized the need for visual aids and non-verbal instruction to ensure patient comprehension.

## Cultural norms and values

Cultural beliefs shaped patient perceptions and rehabilitation adherence. Some patients prioritized traditional healing methods over therapy, leading to low engagement in TOT.

• **Negative Case:** *"Many families believe rest and herbal treatments are better than exercise-based rehabilitation."* (P12)

However, integrating **family members into therapy sessions** improved adherence.

• **Facilitator:** *"We train caregivers to assist with exercises, which increases compliance."* (P8)

**Healthcare system factors**

Limited rehabilitation infrastructure, urban-centric services, and lack of insurance coverage restricted patient access to TOT.

• **Barrier:** *"Patients from smaller cities struggle to access physiotherapy services."* (P4)

• Barrier: *"Many discontinue therapy because they can't afford long-term sessions."* (P7)

 Physiotherapists adapted by developing low-cost home-based TOT programs.

• **Facilitator:** *"We create home-based exercises for patients who can't afford regular visits."* (P17)

**Socioeconomic factors**

Financial constraints affected patient engagement and adherence to TOT.

• **Barrier:** *"Patients often skip sessions because they can't afford to miss work."* (P15)

To address this, physiotherapists provided cost-effective modifications.

• **Facilitator:** *"We teach exercises that require no equipment for home practice."* (P10)

**Language and communication**

Language barriers affected patient understanding and compliance with therapy.

• **Barrier:** *"Patients from rural areas struggle with Urdu instructions."* (P19)

Physiotherapists used non-verbal communication and visual aids to bridge gaps.

• **Facilitator:** *"Demonstrations and gestures help patients follow instructions."* (P9)

## Discussion

This study explored the implementation of TOT by Pakistani physiotherapists and identified six core themes: (1) universal adoption of TOT for motor and balance recovery, (2) underutilization of cognitive rehabilitation, (3) structured approaches to balance training, (4) key barriers including resource and time constraints as well as knowledge gaps, (5) patient-related challenges such as fear and motivation, and (6) cultural and contextual influences spanning family beliefs, healthcare infrastructure, socioeconomic factors, and communication barriers. The findings align with prior research on TOT's effectiveness [5] but highlight context-specific challenges in low-resource settings.

Findings reinforce that TOT is widely accepted as a patient-centered, functional rehabilitation approach in stroke care, aligning with existing research [27]. Participants emphasized that real-life task integration, rather than isolated exercises, enhances motor recovery and neuroplasticity. However, despite recognizing TOT's benefits, variability in implementation was observed, influenced by institutional resources, therapist expertise, and patient adherence. Prior studies in high-income countries report standardized TOT frameworks [18,19], whereas in Pakistan, therapists adapted TOT based on available resources and cultural norms. This highlights the need for locally adapted TOT protocols that maintain fidelity while accounting for contextual constraints.

Physiotherapists structured motor function training into upper and lower limb rehabilitation, focusing on gross motor (e.g., reaching, stepping) and fine motor tasks (e.g., grip, manipulation). These approaches align with task-specific motor learning theories [5,27]. However, two critical barriers emerged, first is patient reluctance due to fear of movement, a phenomenon widely documented in post-stroke rehabilitation. [28,29] The second is limited access to specialized rehabilitation equipment, particularly in under-resourced settings which is also reported in literature as barrier for effective rehabilitation. [30] Therapists modified exercises using low-cost household items (e.g., stacking objects, chair-assisted gait training), similar to rehabilitation models in other LMICs [31].

A significant finding was that TOT is primarily applied for motor rehabilitation, with limited integration of cognitive training. While research supports the inclusion of dual-task exercises for cognitive recovery [32], therapists in this study rarely incorporated structured cognitive elements into their practice. Therapists reported that main barrier is Lack of therapist training in cognitive rehabilitation. Some therapists reported difficulty engaging cognitively impaired patients, which led to therapy non-adherence. A few therapists used simple cognitive tasks (e.g., problem-solving while walking, object recall games) to enhance engagement.

Findings confirm the importance of balance training in stroke rehabilitation, in line with existing neuro-rehabilitation research. [33] Participants used static and dynamic exercises (e.g., single-leg stance, obstacle navigation) to improve stability. And they have reported the barrier of limited availability of balance-training equipment (e.g., balance boards, VR-based therapy), a challenge also reported in LMIC rehabilitation studies. [34] In current study therapists reported that some clinics lacked adequate space, forcing therapists to improvise exercises using minimal equipment. And few therapists reported that terrain-based gait training and weight-shifting drills were used as cost-effective alternatives.

Across all therapy domains, therapists identified three primary barriers to TOT implementation first is Resource and Time Constraints – Therapists faced heavy patient caseloads and insufficient access to specialized rehabilitation tools, limiting therapy frequency and quality. While some therapist prioritized functional tasks over isolated exercises to optimize session time. Second barrier was patient-related factors (Motivation and Fear of Movement) – Cognitive impairments, low motivation, and financial limitations affected therapy adherence, consistent with prior LMIC studies. [35,36] Physiotherapists in current study reported that they used goal-setting, caregiver involvement, and culturally relevant patient education to improve compliance. Third barrier is lack of referral pathways and financial support for rehabilitation restricted long-term engagement. In response to this barrier some therapists developed home-based TOT programs to ensure continuity of care.

Findings reveal how cultural, economic, and systemic factors shape rehabilitation practices in Pakistan, emphasizing context-driven modifications. According to therapists some patients preferred traditional healing over physiotherapy, leading to delayed engagement. However, a qualitative study reported that family integration into therapy improved adherence [37] which aligns well with suggestion given by few therapist in current study. Financial Barriers and Healthcare Access – Uninsured rehabilitation services and high out-of-pocket costs limited therapy continuation, similar to other LMIC rehabilitation literature. [38] Language Barriers and Communication Challenges – Therapists used non-verbal cues and demonstrations to accommodate patients from diverse linguistic backgrounds [39].

## Conclusion

This study highlights how Pakistani physiotherapists implement TOT in stroke rehabilitation, emphasizing its role in motor function and balance recovery, while cognitive rehabilitation remains underutilized due to training gaps and patient engagement challenges. Key barriers include resource constraints, time limitations, and cultural influences, which therapists address through home-based adaptations and caregiver involvement. To enhance TOT effectiveness, standardized clinical guidelines, cognitive rehabilitation training, and policy reforms are needed. Future research should explore cost-effective, scalable rehabilitation models to improve stroke recovery outcomes in resource-limited settings.

## Impact and implications statement

- This study highlights how Pakistani physiotherapists adapt TOT in stroke rehabilitation, addressing barriers such as limited infrastructure, financial constraints, and cultural influences in low-resource settings.

- Findings reveal the underutilization of cognitive rehabilitation within TOT, emphasizing the need for integrating cognitive training into physiotherapy education and clinical practice to enhance patient outcomes.

- The study calls for standardized TOT protocols tailored to resource-limited settings and policy reforms to improve access to rehabilitation services, including financial support and structured therapist training programs.

## Supporting information

**S1. Supporting information questionnaire.**
(DOCX)

**S2. Supporting information data file.**
(DOCX)

**S3. Supporting information.**
(DOCX)

## Author contributions

**Conceptualization:** Ayesha Afridi, Arshad Nawaz Malik, Fajar Shahid, Aneeqa Iftikhar, Maryam Rehman.

**Data curation:** Ayesha Afridi.

**Formal analysis:** Neha Raheel.

**Investigation:** Neha Raheel, Fajar Shahid.

**Methodology:** Ayesha Afridi, Neha Raheel, Umme Aiman, Fajar Shahid.

**Project administration:** Ayesha Afridi, Arshad Nawaz Malik.

**Software:** Ayesha Afridi, Arshad Nawaz Malik, Umme Aiman, Aneeqa Iftikhar, Maryam Rehman.

**Supervision:** Arshad Nawaz Malik.

**Validation:** Umme Aiman, Maryam Rehman.

**Visualization:** Arshad Nawaz Malik, Maryam Rehman.

**Writing – original draft:** Arshad Nawaz Malik, Umme Aiman, Fajar Shahid, Aneeqa Iftikhar, Maryam Rehman.

**Writing – review & editing:** Arshad Nawaz Malik, Umme Aiman, Aneeqa Iftikhar.

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
