## [Decision Letter · Decision Letter 0]

25 Apr 2025

PONE-D-25-13944‘Task-Oriented Training in Stroke Rehabilitation: Qualitative study on Perspectives and Challenges among Pakistani Physiotherapists’PLOS ONE

Dear Dr. Afridi,

Thank you for submitting your manuscript to PLOS ONE. After careful consideration, we feel that it has merit but does not fully meet PLOS ONE’s publication criteria as it currently stands. Therefore, we invite you to submit a revised version of the manuscript that addresses the points raised during the review process.

We look forward to receiving your revised manuscript.

Kind regards,

Nadinne Alexandra Roman, Ph.D.

Academic Editor

PLOS ONE

Reviewers' comments:

Reviewer's Responses to Questions

**Comments to the Author**

1. Is the manuscript technically sound, and do the data support the conclusions?

Reviewer #1: Partly

Reviewer #2: Yes

2. Has the statistical analysis been performed appropriately and rigorously? 

Reviewer #1: I Don't Know

Reviewer #2: Yes

3. Have the authors made all data underlying the findings in their manuscript fully available?

Reviewer #1: No

Reviewer #2: Yes

4. Is the manuscript presented in an intelligible fashion and written in standard English?

Reviewer #1: Yes

Reviewer #2: Yes

5. Review Comments to the Author

Reviewer #1: The manuscript presents valuable insights into the use of Task-Oriented Training (TOT) in stroke rehabilitation among physiotherapists in Pakistan. However, there are several areas that require clarification, restructuring, and additional evidence to strengthen the study's scientific rigor and presentation.

Abstract:

• Abbreviate Task-Oriented Training as TOT upon its first mention and use the abbreviation consistently throughout the manuscript.

• The results section of the abstract may include numerical values derived from the data analysis (e.g., percentage of therapists using TOT, common barriers identified, response rates) as applicable.

Introduction:

• Replace vague expressions like “with the rising incidence of stroke in LMICs” with quantitative epidemiological data, citing specific incidence or prevalence rates.

• The statement “influences survivors' ability to resume work” should specify the age group most affected, supported by relevant literature.

• Clarify or provide citation evidence for the following claim:

“However, while TOT has been extensively considered in Western contexts of rehabilitation, its application in LMICs, particularly in Pakistan, remains underexplored.”

Include region-specific studies or a literature gap analysis to support this assertion.

References:

• Reference no. 10 is not accessible—please provide a valid source or replace with an alternative that is publicly available.

• Ensure all citations used to support methodology or clinical practice (especially those describing rehabilitation procedures) are appropriately cited.

Methodology:

• Sample Size: Justify why only 22 therapists were selected for purposive sampling. Explain whether saturation was achieved.

• Questionnaire Validation: Clearly state:

o How many therapists were involved in validating the semi-structured interview tool?

o Their qualifications and professional background.

o Whether these therapists had prior knowledge or experience with TOT (as part of inclusion criteria).

• Language of Questionnaire: Specify the language in which the questionnaire was developed and administered.

• Content Validation: Confirm if the tool explicitly assesses whether the participants are acquainted with the TOT approach.

Data Analysis:

• Report response rates across different sections/domains of the semi-structured questionnaire.

• Include any correlation or association found between participant characteristics (e.g., experience level, clinical setting) and the use or understanding of TOT.

• Ensure the similarity index of the manuscript is <10% and revise any plagiarized or overly similar content from prior publications.

Results:

• Present all findings using numerical values, e.g.,

o “68% of therapists use TOT in upper limb rehabilitation”

o “Only 27% of therapists incorporate cognitive tasks into sessions”

• When reporting themes from qualitative analysis, quantify the responses where possible (e.g., “12 out of 22 participants identified lack of training as a barrier”).

Discussion:

• Begin the discussion section with a summary of key findings.

• Compare your findings with existing studies or systematic reviews, particularly from LMICs or similar rehabilitation settings.

• Offer explanations for any discrepancies between your findings and prior research.

• Discuss the strength of the evidence supporting TOT’s effectiveness, referencing high-quality studies (e.g., RCTs, meta-analyses) that support its use in stroke rehabilitation.

Reviewer #2: Strengths:

The use of a qualitative descriptive methodology with Braun & Clarke's six-step thematic analysis is appropriate and rigorously applied.

The study addresses a significant gap in rehabilitation literature from low-resource settings.

Ethical considerations, participant diversity, and reflexivity practices are commendable.

Results are well-structured into meaningful themes that reflect both practice and systemic realities.

Suggestions for Improvement:

Consider slightly condensing some repetitive statements in the introduction and discussion for improved readability.

Some grammatical and syntax refinements (minor) are recommended; for instance, changing “has extensively reported” to “have extensively reported” (line 91) for subject-verb agreement.

A brief table summarizing key themes and subthemes with representative quotes would enhance the clarity of findings.

Overall, this manuscript presents valuable insights into the real-world implementation of TOT in a low-resource setting and has practical implications for education, policy, and future research. I recommend acceptance with minor language editing.

6. PLOS authors have the option to publish the peer review history of their article (what does this mean? ). If published, this will include your full peer review and any attached files.

**Do you want your identity to be public for this peer review?** For information about this choice, including consent withdrawal, please see our Privacy Policy .

Reviewer #1: No

Reviewer #2: No

---

## [Author Response · Author response to Decision Letter 1]

29 Apr 2025

We sincerely thank the reviewers for their careful reading and constructive feedback. In response, we have thoroughly revised the manuscript to incorporate all suggestions, including clarifying our methodology, enhancing quantitative reporting, updating and expanding our literature support, streamlining repetitive text, improving grammatical precision, and adding new tables and supporting data files. We believe these changes have significantly strengthened the clarity, rigor, and transparency of our work, and we look forward to any further guidance.

---

## [Decision Letter · Decision Letter 1]

18 Jun 2025

PONE-D-25-13944R1‘Task-Oriented Training in Stroke Rehabilitation: Qualitative study on Perspectives and Challenges among Pakistani Physiotherapists’PLOS ONE

Dear Dr. Afridi,

Thank you for submitting your manuscript to PLOS ONE. After careful consideration, we feel that it has merit but does not fully meet PLOS ONE’s publication criteria as it currently stands. Therefore, we invite you to submit a revised version of the manuscript that addresses the points raised during the review process.

We look forward to receiving your revised manuscript.

Kind regards,

Nadinne Alexandra Roman, Ph.D.

Academic Editor

PLOS ONE

Journal Requirements:

Reviewers' comments:

Reviewer's Responses to Questions

**Comments to the Author**

1. If the authors have adequately addressed your comments raised in a previous round of review and you feel that this manuscript is now acceptable for publication, you may indicate that here to bypass the “Comments to the Author” section, enter your conflict of interest statement in the “Confidential to Editor” section, and submit your "Accept" recommendation.

Reviewer #1: All comments have been addressed

Reviewer #2: All comments have been addressed

2. Is the manuscript technically sound, and do the data support the conclusions?

Reviewer #1: Yes

Reviewer #2: Yes

3. Has the statistical analysis been performed appropriately and rigorously? 

Reviewer #1: I Don't Know

Reviewer #2: Yes

4. Have the authors made all data underlying the findings in their manuscript fully available?

Reviewer #1: Yes

Reviewer #2: Yes

5. Is the manuscript presented in an intelligible fashion and written in standard English?

Reviewer #1: Yes

Reviewer #2: Yes

6. Review Comments to the Author

Reviewer #1: The description of the sampling method is generally clear; however, a few points require clarification and elaboration. First, while purposive sampling is for targeting experienced professionals, the rationale behind selecting only physiotherapists with a Master’s degree in Neuro-Muscular Physical Therapy should be explained more explicitly. Clarifying whether this criterion was intended to ensure a homogeneous expert group or to reflect a standard qualification in the region would strengthen the methodological justification. Is it intended in inclusion criteria??

Additionally, while the recruitment sources are diverse, it would be helpful to discuss any potential selection biases that may arise from relying on professional associations and academic networks. Please comment on whether this urban and semi-urban focus was intentional and how it may affect the generalizability of findings.

Finally, consider providing brief information on how many participants were expected from each source for sample composition.

Reviewer #2: The content sounds good and appreciating your efforts for revision. The discussion and methodology are approriate

7. PLOS authors have the option to publish the peer review history of their article (what does this mean? ). If published, this will include your full peer review and any attached files.

**Do you want your identity to be public for this peer review?** For information about this choice, including consent withdrawal, please see our Privacy Policy .

Reviewer #1: No

Reviewer #2: **Yes: ** Sridhar Arumugam

---

## [Author Response · Author response to Decision Letter 2]

24 Jun 2025

Response to Reviewers

• Journal Requirements:

Response: We have thoroughly reviewed all references in our manuscript. None of the cited sources are currently listed as retracted, based on searches in PubMed and the Retraction Watch Database as of June 2025. Additionally, missing reference details have been completed, and formatting was corrected to conform to journal style guidelines.

• Reviewer #1: The description of the sampling method is generally clear; however, a few points require clarification and elaboration. First, while purposive sampling is for targeting experienced professionals, the rationale behind selecting only physiotherapists with a Master’s degree in Neuro-Muscular Physical Therapy should be explained more explicitly. Clarifying whether this criterion was intended to ensure a homogeneous expert group or to reflect a standard qualification in the region would strengthen the methodological justification. Is it intended in inclusion criteria??

Response: Thank you for this important observation. We have now clarified in the Materials and Methods section (under “Sample Size and Participants Characteristics”) that the selection criterion was intended to ensure a homogeneous expert group with specialized training in neurorehabilitation. This degree represents the standard qualification for clinical specialization in stroke rehabilitation in the Pakistani context.

• Additionally, while the recruitment sources are diverse, it would be helpful to discuss any potential selection biases that may arise from relying on professional associations and academic networks. Please comment on whether this urban and semi-urban focus was intentional and how it may affect the generalizability of findings.

Response: We thank the reviewer for this insightful observation. We acknowledge the potential for selection bias arising from our recruitment strategy, which relied on professional associations and academic networks. This approach may have limited participation from therapists working in rural or informal settings. However, the focus on urban and semi-urban participants was intentional, as these areas represent the principal hubs for structured neurorehabilitation services in Pakistan. Consequently, the findings are most representative of physiotherapists operating in formal clinical environments where task-oriented training is more feasibly implemented.

• Reviewer #1 suggested including brief information on how many participants were expected from each recruitment source to clarify sample composition.

Response: We appreciate this suggestion. While we did not set predefined quotas for recruitment sources, participants were drawn proportionally based on availability and responsiveness from each channel. Approximately 40% of the participants were recruited through professional physiotherapy associations, 35% through academic networks, and 25% via direct referrals from rehabilitation centers. This distribution reflects the practical accessibility of experienced neurorehabilitation professionals within the urban and semi-urban clinical ecosystem.

• Reviewer #2 commended the manuscript for its content, discussion, and methodological rigor.

Response: We sincerely thank Reviewer #2 for the encouraging feedback and appreciation of the revisions. We are pleased that the methodological clarity and depth of discussion meet the expectations and contribute meaningfully to the literature on task-oriented training in stroke rehabilitation.

---

## [Decision Letter · Decision Letter 2]

5 Aug 2025

‘Task-Oriented Training in Stroke Rehabilitation: Qualitative study on Perspectives and Challenges among Pakistani Physiotherapists’

PONE-D-25-13944R2

Dear Dr. Afridi,

We’re pleased to inform you that your manuscript has been judged scientifically suitable for publication and will be formally accepted for publication once it meets all outstanding technical requirements.

Kind regards,

Nadinne Alexandra Roman, Ph.D.

Academic Editor

PLOS ONE

Additional Editor Comments (optional):

Reviewers' comments:

Reviewer's Responses to Questions

**Comments to the Author**

1. If the authors have adequately addressed your comments raised in a previous round of review and you feel that this manuscript is now acceptable for publication, you may indicate that here to bypass the “Comments to the Author” section, enter your conflict of interest statement in the “Confidential to Editor” section, and submit your "Accept" recommendation.

Reviewer #1: All comments have been addressed

Reviewer #2: All comments have been addressed

2. Is the manuscript technically sound, and do the data support the conclusions?

Reviewer #1: Yes

Reviewer #2: Yes

3. Has the statistical analysis been performed appropriately and rigorously? 

Reviewer #1: I Don't Know

Reviewer #2: Yes

4. Have the authors made all data underlying the findings in their manuscript fully available?

Reviewer #1: Yes

Reviewer #2: Yes

5. Is the manuscript presented in an intelligible fashion and written in standard English?

Reviewer #1: Yes

Reviewer #2: Yes

6. Review Comments to the Author

Reviewer #1: The description of the methodology in the abstract could be more detailed to give readers a clearer understanding of the study’s design and approach. Additionally, it would strengthen the manuscript if the authors briefly mentioned the key strengths and limitations of their study

Reviewer #2: The revised manuscript titled “Task-Oriented Training in Stroke Rehabilitation: Qualitative Study on Perspectives and Challenges among Pakistani Physiotherapists” provides valuable insights into the implementation of task-oriented training (TOT) in low-resource contexts. The authors have addressed previous reviewer comments thoroughly and have enhanced the methodological rigor of the manuscript. Key strengths of this study include:

Clear justification for using qualitative descriptive methodology.

Thoughtful sampling of experienced neurorehabilitation physiotherapists.

Systematic thematic analysis using Braun and Clarke's six-step framework.

Culturally grounded insights that address practical barriers to implementing TOT in Pakistan.

The addition of detailed explanations regarding the purposive sampling strategy (focused on master's-level neurorehabilitation-trained physiotherapists) and recruitment from urban/semi-urban settings provides useful context. The authors' acknowledgment of limitations related to rural underrepresentation is appreciated.

The themes identified are well-developed and aligned with existing literature. Importantly, the authors go beyond surface-level findings and present nuanced views on cognitive rehabilitation and cultural barriers.

7. PLOS authors have the option to publish the peer review history of their article (what does this mean? ). If published, this will include your full peer review and any attached files.

**Do you want your identity to be public for this peer review?** For information about this choice, including consent withdrawal, please see our Privacy Policy .

Reviewer #1: No

Reviewer #2: **Yes: ** SRIDHAR ARUMUGAM

---

## [Editor Report · Acceptance letter]

PONE-D-25-13944R2

PLOS ONE

Dear Dr. Afridi,

I'm pleased to inform you that your manuscript has been deemed suitable for publication in PLOS ONE. Congratulations! Your manuscript is now being handed over to our production team.

Kind regards,

on behalf of

Dr. Nadinne Alexandra Roman

Academic Editor

PLOS ONE